# Exploring the Potential of Machine Learning Algorithms Associated with the Use of Inertial Sensors for Goat Kidding Detection

**DOI:** 10.3390/ani14060938

**Published:** 2024-03-19

**Authors:** Pedro Gonçalves, Maria do Rosário Marques, Ana Teresa Belo, António Monteiro, João Morais, Ivo Riegel, Fernando Braz

**Affiliations:** 1Instituto de Telecomunicações, Escola Superior de Tecnologia e Gestão de Águeda, Universidade de Aveiro, 3830-193 Aveiro, Portugal; 2Instituto Nacional de Investigação Agrária e Veterinária I.P. (INIAV), Avenida Professor Vaz Portugal, 2005-424 Vale de Santarém, Portugal; rosario.marques@iniav.pt (M.d.R.M.); anateresa.belo@iniav.pt (A.T.B.); 3Escola Superior Agrária do Instituto Politécnico de Viseu e CERNAS, Centro de Recursos Naturais, Ambiente e Sociedade, Quinta da Alagoa, Estrada de Nelas, 3500-606 Viseu, Portugal; amonteiro@esav.ipv.pt; 4Instituto de Telecomunicações, Departamento de Eletrónica Telecomunicações e Informática, Universidade de Aveiro, 3830-193 Aveiro, Portugal; antoniojoao10@ua.pt; 5Instituto Federal Catarinense, Campus Araquari, Araquari 89245-000, Brazil; ivo.riegel@ifc.edu.br (I.R.); fernando.braz@ifc.edu.br (F.B.)

**Keywords:** goat kidding detection, inertial sensors, stream learning, concept drift, edge computing, precision livestock farming

## Abstract

**Simple Summary:**

Automatic detection of births allows timely assistance, protecting offspring and mothers, without requiring continuous human surveillance. A mechanism based on Machine Learning was developed using wearable inertial sensors, enabled with real-time communication. This mechanism runs on a minicomputer housed in livestock facilities and uses inertial data classification to detect and notify the human operator of goat kidding events. Preliminary results demonstrate behavior changes four hours before kidding and allow for the identification of the kidding hour with an accuracy of 61%.

**Abstract:**

The autonomous identification of animal births has a significant added value, since it enables for a prompt timely human intervention in the process, protecting the young and the mothers’ health, without requiring continuous human surveillance. Wearable inertial sensors have been employed for a variety of animal monitoring applications, thanks to their low cost and the fact that they allow less invasive monitoring process. Alarms triggered by the occurrence of events must be generated close to the events to avoid delays caused by communication latency, which is why this type of mechanism is typically implemented at the network’s edge and integrated with existing auxiliary mechanisms on the Internet. Although the detection of births in cattle has been carried out commercially for some years, there is no solution for small ruminants, especially goats, where the literature does not even report any attempts. The current work consisted of a first attempt at developing an automatic birth monitor using inertial sensing, as well as detection techniques based on Machine Learning, implemented in a network edge device to assure real-time alarm triggering. Thus, two concept drift detection techniques and seven kidding detection mechanisms were developed using data classification models. The work also includes the testing and comparison of learning results, both in terms of accuracy and of computational costs of the detection module, for algorithms implemented. The results revealed that, despite their simplicity, concept drift algorithms do not allow kidding detection, whereas classification-algorithm-based static learning models do, despite the unbalanced character of the dataset and its reduced size. The learning findings are quite promising in terms of computational cost and its suitability for deployment on edge devices. The algorithm demonstrates behavior changes four hours before kidding and allows for the identification of the kidding hour with an accuracy of 61%, as well as the capacity to improve the overall learning process with a larger dataset.

## 1. Introduction

Usually, kidding occurs without assistance, but when kids are not presented in the right position or are too large to pass the birth canal, producers or, in extreme cases, veterinary assistance may be required. Thus, permanent human monitoring is necessary. This is demanding of time and economically costly, and so far, a thankless task as there is no precise prediction for time of kidding. The spread of wearable sensors, Internet of Things (IoT), and Artificial Intelligence (AI) technologies has been facilitating animal behavior and birth surveillance using IoT-like monitoring devices [1] and deep learning techniques [2]. Those technologies have succeeded in determining behavior patterns and monitoring activity, and have been shown to be very promising tools to improve the decision-making process in livestock management, as they theoretically can identify the set of typical behaviors observed during lambing and kidding referred to in the literature [3,4]. So, the development of an autonomous tool that can detect the onset of the process is of enormous value, both in economic and animal welfare terms.

Neonatal mortality of ruminant offspring is a huge concern, with up to 50% of all pre-weaning deaths occurring on the days 1–3 of live in sheep [5]. In goats, estimates of kid mortality range from 7.8% [6] to 37% [7], and in commercial dairy goat farms, in New Zealand, it was reported that 90% of mortality occurred before weaning, with a higher frequency in early life [8]. Goats display metabolic, behavioral, and physiological changes indicating kidding time is approaching [9]. Vas et al. [4] reported that goats, at the end of the pregnancy, moved further distances than in the first or second third of gestation. They also reported that, in late pregnancy, goats move away from the feeders and other animals. These results can indicate that goats that are close to parturition have higher spatial needs compared to earlier phases of pregnancy and can show how personal space needs are flexible even throughout an individual’s life. At the end of gestation, goats spent a large amount of time eating, even at resting time, but hardly during the afternoon feeding. Goats had increased resting and decreased feeding time during the morning observations at the end of gestation, but the frequency of resting and feeding behavior remained unchanged at noon [4].

### 1.1. Sensory for Parturition Detection

Over the last decade, substantial work has been conducted on automatic parturition detection, both by academics [10,11] and companies [12,13,14], mainly on large ruminants. Various technologies, including sensors (inclinometers, accelerometers, pedometers), wearable devices (abdominal belts, vaginal probes, and devices placed in the vagina or on the vulvar lips), imaging technology, and connected devices monitoring physiological parameters and behavior, have been used for calving detection in dairy cows (reviewed by Saint-Dizier and Chastant-Maillard [15]).

Less invasive wearable devices, such as activity sensors, are now in use in dairy herds, most of them to assist reproductive management (e.g., estrous, mating, calving). They are technologies based on activity-recording tri-axial accelerometers worn as an ear tag [16], nose halter, neck collar [17], or leg (preferably rear) pedometer [18]. However, no commercial references to items for detecting parturition in small ruminants were found, and the literature analysis revealed a small number of works on the detection of parturition in sheep [19,20,21,22,23,24,25,26], but regarding goats, just a dataset created throughout the monitoring of kidding goats [26].

Smith et al. [23] studied the behavior of 76 pregnant ewes fitted with a three-axis enabled collar system for up to 17 days, recording the births during the field trials with day and night vision cameras to detect parturition time. They analyzed the changes in the ewes’ activity and measured it across time by computing the distance between an activity distribution and baseline activity. The results showed that the best-performing algorithm variant had a Mean Absolute Error (MAE) of 5.33 h between the estimated and human-registered parturition occurrence.

Williams et al. [26] studied the lying behavior of ninety-six Bluefaced Leicester × Welsh Mountain crossbred ewes managed to lamb indoors, and 80 Welsh Mountain ewes managed to lamb during grazing, gathering the acceleration values at 1 min intervals for at least 14 d prior to parturition using a LED-mounted data logger. Video equipment was also used to identify lambing and to assess lying predictions of total lying time, mean lying-bout duration, and total number of lying bouts. This study successfully validated data loggers in lying behavior identification for both the housed and for the grazing flocks. Dobos et al. [19] studied the pre-lambing behavioral changes of 20 pregnant Merino grazing ewes using GNSS devices assessing the mean hourly speed seven days before to seven days after lambing, the mean hourly speed before and after lambing, and the distance between lambing ewes during the lambing process. Their study reported a dramatic decrease in the mean daily speed, as previously suggested by Broster et al. [27], since ewes prolonged their stay at the lambing site, near the new-born lamb, due to its lower mobility [28].

Zobel et al. studied goat lying behavior [29] using rear left leg data loggers enabled with inertial sensors in video-equipped pens. They concluded that the loggers could efficiently record the lying behavior in both mature, pregnant does and younger goats and identified the left- and right-side lying position of the animal, despite reporting problems associated with the rotation of the sensor around the leg. Zobel et al. [18] monitored 420 dairy goats’ lying behavior with the same data logger to study the relationship between the dry period and negative energy balance, and to determine if lying behavior changes are indicative of goats’ metabolic status. It was also observed that healthy does showed increasing activity as parturition neared, and dramatic drops in lying times in the day before, same day, and day after kidding, and that does reduced lying time on the day of kidding. 

Fogarty et al. produced extensive work detecting sheep lambing using inertial loggers [20,21,30] attached to the ear tags. They analyzed the activity and identified a set of hectograms relating to animal behavior during lambing process. By correlating the behaviors before, during, and after the event, the authors concluded that although behaviors varied throughout the birth process, the variation between sheep and cattle did not allow them to be used as indicators of the occurrence of birth [20]. In a later work, they developed a model that includes GNSS, ear tag accelerometer, and weather data, which detected 84% of the parturition dates and showed the feasibility of lambing prevision 12 h before the parturition events.

According to the literature, there is currently no study that can accurately detect kidding events in real time, nor have any reports of goat birth detection attempts been made. 

### 1.2. Data Stream Analysis

A typical solution for the collar-originated data stream classification consists of the use of concept drift-based mechanisms [31,32,33,34] to detect and adapt the learning model to the change in the stream [35]. As such, the present work documents the development of an automatic tool for the detection of goat kidding process based on a concept drift approach. Although it was used as a public dataset, the possibility of running several concept drift detection algorithms in the monitoring gateway, as well as the times needed to detect the concept drift associated with the kidding process, was verified [35]. Compared to static datasets, data streams pose new challenges for ML and data mining, since static dataset data mining methods are not able to efficiently analyze the fast-growing amount of data because of limited computational resources, such as memory and time to classify, the inability to supervise the streamed data in a timely manner, and especially because sometimes the underlying model changes, which cannot be detected through a static learning model [36,37].

Data streams, such as the animal monitoring data stream [38,39], evolve over time in a stream of non-stationary distribution data and infinite size [40], as a function of the evolution of the monitored magnitude, such as the pitch angle of a goat’s neck. Sometimes animal behavior changes, for example because labor is starting, and this behavior cannot be modelled through the learning process carried out earlier, with the data obtained during the period before kidding. The change in data characteristics due to a change in behavior is called concept [33], and its learning is performed through a set of different learning algorithms [41].

Changes in concept are usually classified as abrupt [41,42] or gradual [43], and detection algorithms [35] detect the points/small intervals where the change occurs and trigger the learning model replacement [41]. Drift detectors monitor various properties of stream data, such as standard deviation, predictive error, instance distribution, or stability [37]. In turn, detectors can be classified as reactive when the detection mechanism triggers the learning model, or proactive when they analyze some drift information, such as the historical drift rate information, before the drift occurs.

Over the past two decades, much work has been conducted on algorithms, techniques, and data analysis platforms for detecting context switching in data streams [44], in new application areas [45], as well as in the definition of metrics for detection benchmarking. Interestingly, there are few applications related to concept drift in the livestock area [46]. AdWin (Adaptive Windowing) [36,47] is a very popular algorithm for detecting concept changes and keeping updated statistics about a data stream. The strategy is to compute a set of statistical measures along a dataset considering a variable size window. The algorithm defines the window size and calculates the statistics for each window, up to the point a drift is detected. At that point, the algorithm drops old data samples and starts to create a new concept window. KSWin [48] is a newer concept detector algorithm based on Kolmogorov–Smirnov (KS) [48] statistical test. As with AdWin, KSWin also implements a sliding window describing actual context and it monitors the new stream data to detect deviations.

Vázquez-Diosdado et al. [46] documented the development of a combined offline and online learning algorithm to address concept drift effect in animal behavior classification. Under their study, they used inertial sensors to monitor 17 sheep for 39 days, using sensing devices running a K-Nearest Neighbors (KNN)-based algorithm, enabled with a low-power wide area (LPWA) radio module component to communicate offline. The cited authors compared the results of the animal behavior classification through an offline KNN algorithm based on a previous learning process [49] with the results of the offline KNN algorithm and of the online learning algorithm. Finally, they evaluated the results of the hybrid algorithm, with online and offline components. Their results demonstrate the inadequacy of the static learning process when classification conditions vary, proving the advantage of a concept drift-based approach, and the advantage of combining the two algorithmic components in improving learning outcomes.

### 1.3. AI on the Edge

Contrary to what happens with concept drift detectors, classification models based on static models require that previous learning has been performed using a large dataset, and that the learning model is subsequently used on the data to be classified. Decades of dedicated research has been devoted to understanding the effectiveness and applicability of static learning models. A Decision Tree (DT) [50], for example, is a popular algorithm that constructs a hierarchical structure comprising decision nodes and leaf nodes, enabling efficient decision-making processes. SVC (C-Support Vector Classification) [51] takes a different approach by utilizing hyperplanes to classify data into distinct categories. Logistic Regression offers a probabilistic framework for predicting binary outcomes, while Random Forest [52] leverages an ensemble to improve predictive accuracy. KNN [51], on the other hand, makes use of proximity measurements to classify new instances, and Naïve Bayes applies Bayes’ theorem to calculate probabilities for classification purposes. Furthermore, the literature review highlights the rising prominence of XGBoost (eXtreme Gradient Boosting) [51,53,54] as a state-of-the-art static learning model. XGBoost is an optimized implementation of gradient boosting, combining weak models to create a powerful ensemble model. This approach has demonstrated remarkable success in a variety of fields, including computer vision, natural language processing, and predictive analytics. Its ability to handle large-scale datasets, feature importance estimation, and efficient parallel computing has made it a key tool for data scientists and researchers.

Overall, the literature review provides a comprehensive overview of the diverse techniques and models employed in static learning. Researchers have diligently explored the strengths and weaknesses of these models, aiming to enhance their accuracy and efficiency across different domains. The findings contribute to the ongoing advancement of static learning methodologies and provide valuable insights for future research and applications. The construction of learning models requires vast computational resources, from the amount of memory to store learning data to the processing capacity, which very often needs specialized hardware that is only available in datacenter environments. The detection of events like kidding must be performed as close to the animals as feasible so that the warning can be sent swiftly and without relying on Internet connectivity.

Edge computing [55,56] is a computational approach that directs computational data, applications, and services away from cloud servers to the edge of a network, offering clear gains in terms of processing latency and jitter [57]. The existing literature provides many examples of the use of edge-based architectures for the deployment of AI-enabled alarm systems [58,59,60,61,62]. As such, it appears to be a very practical technique for adapting the sensor network [63], decreasing the latency associated with data offloading to the cloud [57,64] to human assistance trigger process.

Thus, the aim of this work was to develop, for the first time, and test an autonomous system for detecting the early stages of goat parturition with the ability to generate alarms that initiate human support for the kidding process when needed. The system that has been developed employs Machine Learning (ML) algorithms over real-time data streams. It was designed to consume low computational resources, to be present at the network’s edge (being tolerant to Internet access disruptions), and to ensure the generation of an effective time-response alarm to the farm’s staff. Several algorithms, including two concept drift (AdWin and KSWin) and learning-based classification algorithms, namely, Decision Trees, SVC, Logistic Regression, Random Forest, KNN, Naïve Bayes, and XGBOOST, allowing the analysis of forecasting process results as well as the computational resources needs of the detection method, were tested.

## 2. Materials and Methods

The prompt detection of births requires that the analysis of the stream data be performed autonomously and close to the animals, allowing it to be completed more quickly and without relying on Internet access. The building of learning models, on the other hand, necessitates a processing capability only available in the Cloud, as well as a massive quantity of data obtained from a set of animals from several flocks; therefore, the algorithms must be trained in the cloud.

### 2.1. Kidding Detecting Architecture

To tackle these varying challenges, the strategy depicted in Figure 1 was devised. This approach involves deploying certain processing capabilities at the edge network, where it evaluates the data stream generated by the collars and detects kidding behavior using a data classification algorithm. Simultaneously, the data stream is transmitted to the cloud, enabling the integration of data from various farms to enhance the development of more resilient learning models.

To take advantage of the sensing platform [63], the instantiation of the edge solution was performed in a previously developed gateway [65] used in a data-gathering trial [38], and it was implemented by a Raspberry PI 3 Model B Plus Rev 1.3. The integration of the edge equipment with the cloud component was carried out using a message broker that communicates via Message Queuing Telemetry Transport (MQTT) [66].

Functional and performance validation was carried out using data from a public dataset [38], with data from animal monitoring during parturition, in which the data were streamed to the system, imitating the functioning of the system under real conditions. Like this, the data were analyzed and the possibility of developing a real-time mechanism that could detect the change in goat behavior during their parturition process though the classification of streamed data was assessed to create an automatic kidding detector.

### 2.2. The Flock and Dataset

Sixteen pregnant Charnequeira (*Capra hircus*) goats and two control non-pregnant goats from National Institute for Agricultural and Veterinary Research (INIAV, Portugal) experimental flock were fit with iFarmTec collars (http://ifarmtec.pt/produtos.html, accessed on 15 February 2024) during the four-week spring kidding season of 2022, between April and May, and birth detail annotations were published with sensory data [38]. Goats grazed during the day and were housed in the shelter at night, keeping the daily routines throughout the period; thus, an ethical statement considering animal welfare does not apply as only data regarding their postal behavior were recorded.

The data structure includes a timestamp, accelerometer (D_x_, D_y_, and D_z_) data, pitch and roll angles, the distance between the collar and the ground as measured by ultrasound, and behavior classification as performed by the collar (i.e., standing, eating, moving, running). The data were periodically sampled at intervals of 10 s and reported from the collars to a central gateway that stored the data [65]. In order to study animal activity using collar sensors, activity indicators (A) based on accelerometer intensity, also known as Dynamic Body Acceleration [67], were defined, and accordingly to [67], the Vectorial sum of Dynamic Body Acceleration (VeDBAtwo), which is best suited for animal movement analysis, was used and defined as shown in Equation (1).
(1)$$A=VeDBAtwo=\sqrt{D_{x}+D_{y}+D_{z}}$$

### 2.3. Edge-Located Drift Detector

The first step towards creating a concept drift mechanism was to count the number of context changes in A, distance, and pitch and roll angle values over the days of kidding. With this effect, the fluctuation of the context on A, distance, and the pitch and roll angles of each collar was carefully examined. The goal was to verify if it is possible to identify differences in behavior as the context changed across the dataset. When the difference between the values for each window was greater than a predefined threshold, the data change was identified. In addition, the computational effort required to determine the context shift was measured.

The dataset data were streamed into the drift concept algorithm to simulate the use of the algorithm in detecting kidding. The Detection Module registered changes in the existing concept in terms of activity changes as a function of the pitch and roll angles. Two data stream analysis algorithms, AdWin [47] and KSWin [48], were used in the data analysis process.

### 2.4. Model-Based Classification Detector

A kidding detection algorithm based on learning models was also performed. The strategy for creating the learning models was based on the generation of multiple classes of data, characterized by the temporal delay between the timestamp of recording the data and the moment of kidding. Several data classes related to the hourly intervals until kidding, lasting one hour, ranging from 5 h before (−5) to one hour after kidding (+1), with 0 corresponding to the period within kidding hour. Table 1 contains the class description, and additional details about dataset can be found in [38].

The data were divided into a training and a verification set, with the training set representing 75% of the samples, and 25% was used to compose the test set and to measure accuracy. Seven ML algorithms were used to build classification models, namely Decision Trees, SVC, Logistic Regression, Random Forest, KNN, Naïve Bayes, and XGBoost.

To assess the computational requirements of the kidding detector, computational requirements of the various detection algorithms were evaluated, specifically in terms of impact on the gateway’s memory and execution time. Additionally, because the gateway connects a set of monitoring devices to the cloud, one for each animal, the size of the flock has a significant impact on the gateway’s computational capacity, which is why a scalability analysis of the process was performed for several animals of the same flock. The tests were carried out on a PC Intel^®^ 11th Gen i7-1165G7@2.80 GHz, 16 GB RAM, and on a Raspberry Pi 3 Model B Plus Rev 1.3 Cortex-A53 1400, 1 GB RAM, both running a Linux Ubuntu 22.04 distribution, with python version 3.9.2.

### 2.5. Detection of the Kidding Hour

In view of the poor performance of kidding prediction results, a new approach to detecting the time of kidding was attempted, bringing together all other classes and trying to train the classification algorithm to distinguish between kidding and non-kidding classes. The reclassification of records worsened the balance of classes, leading to 99.85% of the records belonging to the non-kidding class and 0.15% of records belonging to the kidding class. The class imbalance impacted the performance of the classification ML model; since the model was trained on imbalanced data, it tended to favor the majority class and struggle to predict the minority class accurately. A model that predicts the majority class for every instance may achieve high accuracy but does not demonstrate its ability to correctly identify the minority class, which is often the critical target of interest. This can result in a lower performance of the classification model.

The subsequent approach to optimize the learning model was twofold: (1) a data-balancing solution was sought and implemented and (2) the impact of each of the features on the learning model was evaluated. Then, the features with the greatest impact were chosen.

To overcome the problems associated with an imbalanced dataset, some key approaches can be considered: Weighted XGBoost, where we assign higher weights to the samples of the minority class during model training. By doing so, the algorithm focuses more on learning from the minority class, leading to improved predictions for that class;Undersampling techniques, such as TomeLinks and EditedNearestNeighbours;Oversampling techniques, such as Synthetic Minority Oversampling Technique (SMOTE) and Borderline-SMOTE;Hybrid Techniques, such as SMOTETomeK and SMOTEENN.

The strategy that was followed to overcome the problems associated with the imbalanced dataset consisted of experimenting with the various techniques, with various weights (5, 100, NeighbourhoodCleaningRule) associated with the XGBoost option being tried, and evaluating the option with the best result.

A feature selection process was carried out using an XGBoost internal function that provides a clear assessment of feature importance. To achieve the best results for this model, the optimal parameters for XGBoost through hyperparameter tuning were found. By systematically exploring different combinations of hyperparameters, the model was fine-tuned to maximize its performance and enhance its effectiveness in handling the dataset. This hyperparameter tuning process enabled the identification of the most suitable configuration for XGBoost, leading to improved results and to a more powerful model. 

Having identified the most important features, their impact on the model’s performance was assessed and the five most impactful were selected. Finally, with the selected features, the model was trained again, and its performance was re-evaluated.

## 3. Results

This section presents a set of results relating to identification of the features that have the most impact on the concept drift detection algorithm, evaluating of the model’s various learning algorithms, and determining the feasibility of implementing the concept drift detector at the edge device. 

Figure 2 depicts goat activity on the day of kidding, where the green lines indicate a change in the behavior, i.e., the concept drift detection time, and the blue area represents the kidding period, as noted in the dataset. When compared to earlier and subsequent periods, there appeared to be a decrease in activity throughout the kidding period. It is possible to perceive an increase in activity throughout the period of kidding when compared with the previous and subsequent periods. In addition, abrupt changes in pitch are clear from the observation of Figure 2. These shifts occurred between 6 (Figure 2, ID 7) to 10 h (Figure 2, ID 14 and ID 17) before kidding, being consistent with the expected behavior observed in the first phase of the kidding process, when the goats’ movements reduce as time of kidding approaches and the animals isolate themselves, start pawing at the ground, and lying down/standing up alternately.

### 3.1. Concept Drift Detection 

The number of changes in concept over time was verified by using the change of concept as a criterion for anticipating the occurrence of kidding, and it was possible to check that, within a few hours of the day of kidding, the number of changes was significantly different from the entire period of the entire eve of the kidding day.

Animal activity (A) was found to be the factor with the biggest changes on the day of birth in the assessment of the most impacting features in the concept drift identification procedure, followed by the pitch angle and, finally, the roll angle. Table 2 shows that there was an increase for all animals in all features analyzed between the day before (E) to the day of parturition (P). All animals consistently had an increase in concept changes on the day of parturition, especially in terms of activity from the eve to the day of parturition, with values ranging from 125% to 200%, which was why the activity for context change detection was employed.

A more detailed analysis of the changes in context across time revealed that the changes were most concentrated in the hour preceding the kidding and at the time slot identified as the time of kidding. Figure 3 and Figure 4 show the contexts detected around the kidding time using a AdWin and KSWin algorithms.

The AdWin algorithm detected a lower average number of contexts across hourly intervals than the KSWin algorithm, analyzing all majors tested, however the number of contexts detected fluctuated from one change 3 h earlier, to an average of 2.5 context changes at the time of kidding. The pattern of context changes decreased progressively in the hourly intervals following the kidding event.

### 3.2. Model-Based Classification Algorithm Performance Comparison

In general, the accuracies of the kidding detection process, as described in Table 3, were very low for all algorithms and data from any of the temporal classes, which may be explained by the limited number of birth events used in the learning process. Furthermore, as the time of kidding approached, the accuracy improved gradually, reaching a maximum for the class corresponding to the time of kidding and increasing immediately for the class corresponding to the later time.

The tests demonstrated that the learning models with the best performance were RF and KNN, which had close accuracies, but that XGBoost also showed an interesting performance. Another intriguing detail of the prediction results, which must be confirmed with additional testing and data, is that RF and KNN alternated in terms of performance across the defined classes.

Table 3 compares data classification models using three metrics: recall, precision, and f1-score. Precision is the percentage of correct model predictions; it measures a classifier’s ability to not categorize an event as positive when it is negative. The recall value indicates the percentage of positive cases identified by the classifier. That is, recall indicates the classifier’s ability to detect all positive events. It is important to understand that a model’s behavior can vary depending on the precision and recall parameters. In this study, a classification model with high recall values reduces the likelihood of false negatives occurring, lowering the risk of goats giving birth without proper monitoring or intervention. The best results were obtained for the Decision Tree model in classes 0 and 5, respectively, at the time of kidding and 5 h before the event.

Figure 5 shows the progression of the f1-score over the defined intervals. The results revealed a clear improvement in prediction in the interval that encompasses the time of kidding, as well as a good understanding of the constraints created by the small number of kidding events in the learning process.

### 3.3. Model-Based Classification Algorithm Cost Comparison

The computational impact of the classification algorithms was evaluated, and as expected, it was lower in PC-based than in Raspberry-based gateways. In terms of the time needed to classify the records (WallTime), as shown in Figure 6, the SVC, Logistic Regression, KNN, Naive Bayes, and XGBoost algorithms performed quite similarly.

In the case of the PC results, there was still a difference in the behavior of the Logistic Regression, KNN, naïve Bayes, and XGBoost algorithms, which took less time than the SVC algorithm, possibly due to differences in task scheduling between the Raspberry PI 32-bit and PC 64-bit versions.

In contrast to the results obtained while assessing classification time, the results obtained in terms of impact on the system memory were more uniform between the various algorithms in general. The results in Figure 7 reveal that KNN, Naive Bayes, and XGBoost had a slightly bigger memory footprint than the others, but the outcomes were similar.

When comparing the Raspberry PI gateway version to the PC, it was possible to notice a much more efficient memory occupation in the Raspberry PI, which is understandable given the Raspberry PI memory constraints, which, despite the efficiency, represented approximately 22% of available memory on the gateway in the case of XGBoost algorithm.

Table 4 summarizes the findings achieved by the various algorithms in terms of precision, detection time, and memory impact resulting from the execution of the kidding detector.

### 3.4. Kidding Hour Detection Results

The baseline of model performance was established using the initial imbalanced dataset, using a RepeatedStratifiedKFold with five folds and two repeats. This cross-validation technique assures that the obtained results are as accurate and reliable as possible by repeatedly stratifying the dataset to create consistent and representative folds for evaluation. To better understand both classes, metrics such as precision, recall, and f1-score were calculated with an average macro. Baseline results are presented in Table 5 and Figure 8. 

The occurrence of kidding events in the dataset was quite low compared to other events or activities. This led to an imbalanced dataset in which the class parturition was underrepresented in comparison to the other classes, accounting for approximately 0.15% of the dataset’s samples. A class imbalance could impact the performance of a classification ML model. Because the model was trained on imbalanced data, it may tend to favor the majority class while struggling to accurately predict the minority class, resulting in lower classification model performance.

After studying classification performance with the unchanged data (Table 5 and Figure 8), it became evident that accuracy and AUC ROC were inadequate metrics for evaluating the model’s performance due to their sensitivity to the imbalanced dataset. The results from these metrics were misleading as they failed to consider the underlying class distribution, leading to an inaccurate representation of the model’s true effectiveness. Similarly, class imbalances could affect AUC ROC, which measures the model’s ability to discriminate between positive and negative cases across various probability thresholds. If the majority class dominates the dataset, the ROC curve may provide overly optimistic performance, hiding the model’s true capability to classify the minority class effectively. To mitigate this, various measures were considered to obtain the optimal model while avoiding misleading conclusions about the chosen model.

Exploring and comparing different techniques (see Figure 9) revealed their impact on the model performance and indicated the most effective strategy for the specific imbalanced dataset under study in terms of accuracy, balanced accuracy, average precision, AUC ROC, recall, f1-score, and Matthews correlation coefficient (MMC) [68].

Observing the results, it was become apparent that while EditedNearestNeighbours did not exhibit the most significant improvements in specific metrics, it did avoid negative impacts on metrics that were noticeably affected by the other techniques. The weighted technique was also worth considering, as it was the fastest to compute despite having slightly lower precision than regular classification. It is important to note that this approach improved recall significantly, which was a crucial aspect of this study because it Is focused on positive events. A higher recall is particularly important since it prioritizes capturing all the positive cases, even if some of them are false positives. This was preferable than encountering false negatives (precision), which might lead to critical consequences, such as missing the goat’s parturition and even putting the mother and the kid’s health and survival at risk. Furthermore, when compared to alternative techniques that had negative impacts on multiple metrics, weighted appeared to be a promising option for achieving well-balanced performance while maintaining computational efficiency. 

To disclose the optimal results using Weighted XGBoost, tests were run with various weights as a model parameter, aiming to identify the weight that provides the best overall results by examining a variety of weights and their corresponding performances. These results, presented in Figure 10, provided a clear visualization of how the model’s performance varied with weight values. This weight value strikes a compromise between various metrics, enhancing the model’s performance while having minimal negative impacts on other metrics. However, employing a weight value of 100 resulted in a higher recall, which is beneficial for capturing positive events. Nonetheless, this comes at the expense of other metrics, which potentially may compromise overall performance. Based on these results, a weight parameter of 5 is recommended as it achieves a more balanced and effective outcome for the model.

After comparing the best results from both the undersampling techniques comparison and the Weighted XGBoost analysis, it was possible to confidently select weighted with a weight parameter of 5 as the preferred technique for this study. It demonstrated an overall superior performance, achieving a balanced trade-off between precision and recall, as evidenced by the f1-score, representing the harmonic mean of these two metrics. Moreover, the MCC highlights the effectiveness of the Weighted 5 approach when compared to the results obtained from the normal dataset. Having solved the issue of dataset imbalance, the effect of the features with the higher impact on the learning process was analyzed using an internal XGBoost internal function, allowing the features with most limited impact to be discarded, and the results are presented in Figure 11 and Figure 12.

Finally, a grid of hyperparameter ranges was created and efficiently sampled to perform K-Fold cross-validation with each combination of values, allowing for computational systematic and efficient exploration of different hyperparameter configurations. Ultimately, at the end of this procedure, the best parameters for the model were identified, allowing its performance to be maximized, delivering the most effective results for the given task.

Table 6 shows the model’s final scores after balancing the dataset and selecting the most impactful features in the learning process—Table 6 indicates the optimal performance achieved. The final parameters identified by the RandomizedSearchCV, which led to the model’s peak scores, were as follows: (i) scale_pos_weight(weight parameter) = 5; (ii) subsample = 0.93; (iii) reg_lambda = 0.000001; (iv) reg_alpha = 20; (v) min_child_weight = 20; (vi) max_depth = 10; (vii) learning_rate = 0.3; (viii) gamma = 0; and colsample_bytree = 0.86.

With the model’s highest scores, the confusion matrix and the report for subsequent analysis were obtained and are presented in Table 7 and Figure 13.

## 4. Discussion

The timely detection of parturition is dependent on an analysis of fast-sensing data, making it suitable to deploy the detector mechanism at the network edge. Regardless of Internet access failures, this approximation guarantees detector responsiveness. Contrary to the computational means traditionally available in the cloud, the computational means at the edge impose specific processing and memory constraints, which are unavailable for data analysis processes.

Although the authors were unaware of the application of concept drift to detect kidding, the data stream analysis technique seemed promising as it could detect changes in goat behavior prior to parturition. Concept drift is a technique for monitoring data streams that is widely used in a variety of scenarios; however, it has proven to be ineffective due to the larger number of context changes detected and the ambiguity of each context’s interpretation. In the current usage scenario, it would constantly trigger the classification model update, which would not be feasible given the gateway’s computational processing restrictions. Furthermore, computational performance findings revealed that AdWin has a computational impact that is like learning-model-based algorithms, both in terms of execution time and used memory.

Because of the small number of kidding samples, tests with algorithms based on learning models show results with low accuracy. However, they are still promising in the sense that by repeating the tests, sampling the sensors at a greater frequency, and monitoring a greater number of animals, we will be able to significantly enrich the learning models and, consequently, the detection outcomes. In general, the Random Forest, KNN, and Naive Bayes algorithms present very similar results, with KNN performing best in the majority of the defined time intervals. Increasing the quantity of training datasets usually improves the performance of the KNN algorithm [69]. Therefore, this trial’s results could greatly improve with further data on kidding behavior. In contrast to the work of Sakai et al. [70], which considers the Decision Tree algorithm as the best when compared to KNN, this work obtained the worst results for the DT algorithm, possibly due to the extremely imbalanced data. Indeed, studies employing this algorithm to predict goat behavior [70] have found that balancing the data improves accuracy.

Benaissa et al. [71] studied a combination of indoor localization and accelerometer sensors to detect calving in 13 pregnant Holstein cows. The results showed that precision, sensitivity, specificity, and overall accuracy declined from 24 to 2 h before calving. In the present study, the kidding detecting precision increased from four hours before kidding (0.3653 for KNN algorithm) to the kidding hour (0.5448 for Random Forest algorithm). The precision was lower than in cows four hours before calving, but improved significantly as the time of kidding approached, reaching values like those obtained for neck accelerometer and localization two hours before calving by Benaissa et al. [71]. At this point, detecting kidding at time intervals as short as four hours to one hour is still difficult; however, changes in behavior observed at ten to six hours (Figure 3) could be used to generate alerts that help assist producers in deciding whether to send animals to the grazing plots or keep them inside because kidding could occur on that day. 

The computational cost of the detection algorithm has a significant impact on the solution’s viability and scalability, particularly given the architectural choice to implement the detection module at the network edge. The SVC, Logistic Regression, KNN, Nave Bayes, and XGBoost algorithms performed similarly in the tested Raspberry Pi gateway, but the Logistic Regression, KNN, Nave Bayes, and XGBoost algorithms showed better performances for time measure results obtained on the PC.

The differences in algorithm performance between the PC and the Raspberry PI-based gateway are perfectly understandable due to the two computers’ different processing capacities. The Raspberry Pi ran the classification algorithms at an average of approximately 790 microseconds, whereas the PC took less than 19 microseconds. In terms of memory, the results looked promising, with the detection algorithms using an average of roughly 190 MB of RAM, representing less than 20% of the Raspberry’s system memory (1 GB). The performance tests revealed that, despite the Raspberry PI gateway’s limited computational capabilities, it will be possible to use it to develop a kidding detector at the network’s edge, with all the advantages of low latency and independence from Internet access.

Compared to classification time results, the impact on the system memory produced more consistent results across algorithms. KNN, Naive Bayes, and XGBoost had a slightly larger memory footprint than the others but generated close results. The comparison of the Raspberry PI gateway version and the PC revealed a much more efficient memory occupation in the Raspberry PI, even though the detection process takes up approximately 23% of the system’s free memory in the case of XGBoost algorithm, demonstrating the need to carry out more detailed tests to determine the scalability of the kidding detection system.

The attempt to detect kidding days yielded results with good accuracy but low recall, which is a common characteristic of unbalanced datasets. So, various strategies for addressing data imbalance were investigated and compared, with the weighted XGBoost technique being selected for its higher performance.

During this process, an analysis of the significance of each feature in the model’s performance revealed that pitch and roll angles, as well as temporal features, ranked among the top five most impactful. This finding corroborated previous research, highlighting the redundancy of linear acceleration components and the limited impact of distance to ground, measured with ultrasound. This research resulted in the selection of the top five features based on their cumulative impact on the learning model.

The learning model’s effectiveness was limited by the size of the dataset, as seen in prior works with larger datasets gathered in sheep [20,21,22,25,72]. Comparing results before and after the data unbalancing process confirmed a trade-off between accuracy and recall, a common outcome in such procedures, as indicated by the initial and final confusion matrix values illustrated in Figure 8 and Figure 13.

Despite these challenges, the preliminary results in both our study and the literature show promise. Therefore, the monitoring process must be repeated with a larger flock and increased sampling frequency to further refine the approach.

## 5. Conclusions

Automated kidding detection is a crucial tool with substantial implications for farm profitability and animal welfare. Concept drift-based algorithms emerge as a highly promising avenue for kidding detection, given that the detection process involves analyzing a data stream—a scenario well-suited for concept drift considerations. However, despite their compatibility with edge devices in terms of computational impact, these algorithms cannot function independently. This is because they detect multiple concept changes during the delivery process without accurately identifying them.

The algorithms based on learning models proved feasible for implementation on a microcomputer in terms of computational demands and execution time, enabling real-time and edge detection.

Additionally, the majority of algorithms took less than 50 microseconds to complete the detection process, allowing alarms to be triggered requesting 696 prompt human assistance when needed.

The effort to identify the goat kidding day resulted in a notable enhancement in the algorithm’s effectiveness, owing to the reduced number of classes the algorithm had to learn. Nevertheless, the outcomes remained unsatisfactory since the detection of kidding day was achieved with only 61% accuracy.

Although the dataset was limited, the results are promising; yet, the low detection procedure accuracy limits their applicability. To confirm the expectation of more convincing accuracy results, new tests with a higher sampling frequency and a larger number of animals undergoing kidding are essential in the future work. Lastly, testing should encompass a scalability assessment to effectively enhance the computing capabilities of the edge element.

## Figures and Tables

**Figure 1 animals-14-00938-f001:**
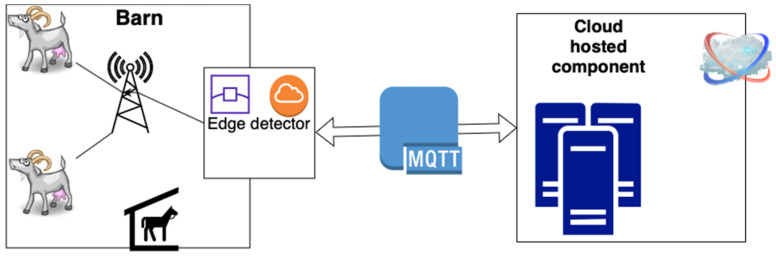
Kidding detector solution architecture.

**Figure 2 animals-14-00938-f002:**
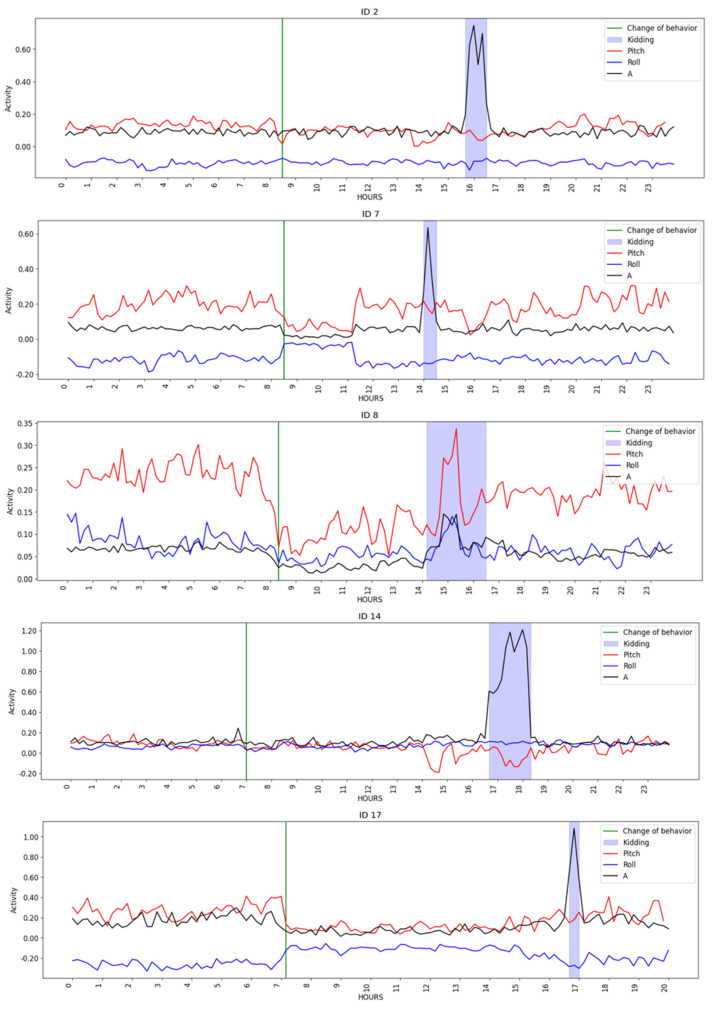
Goats’ activity changes in the period around the kidding time.

**Figure 3 animals-14-00938-f003:**
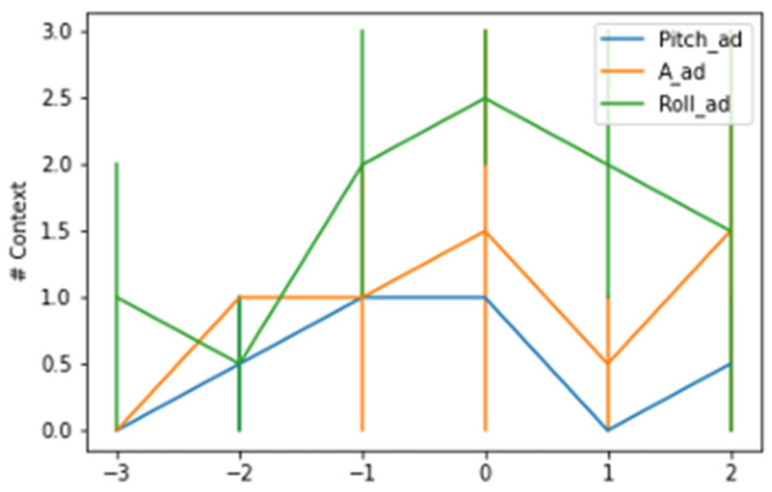
Context changes detected by AdWin algorithm around the hour of kidding.

**Figure 4 animals-14-00938-f004:**
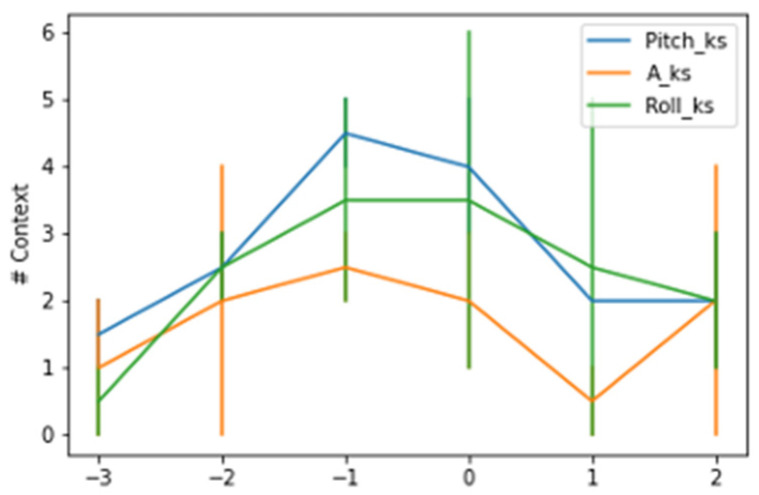
Context changes detected by KSWin around the hour of kidding.

**Figure 5 animals-14-00938-f005:**
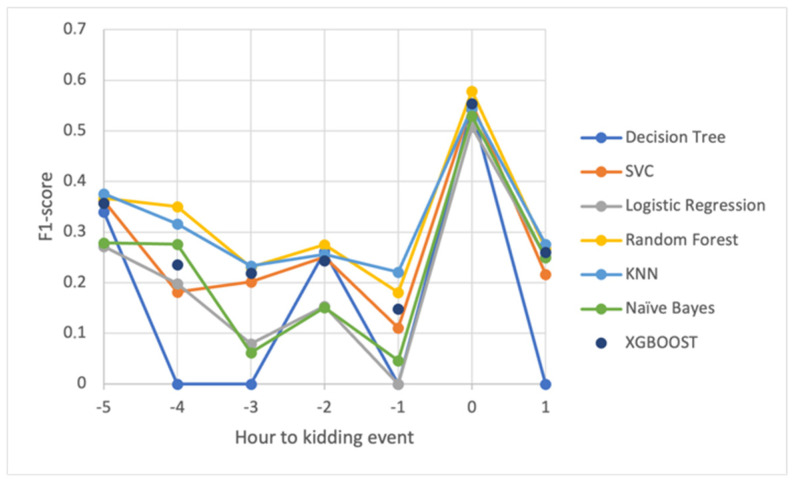
The f1-scores (the harmonic average of sensitivity and precision) were higher at kidding event hour for all classification algorithms. Note: SVC—C-Support Vector Classification; KNN—K-Nearest Neighbors.

**Figure 6 animals-14-00938-f006:**
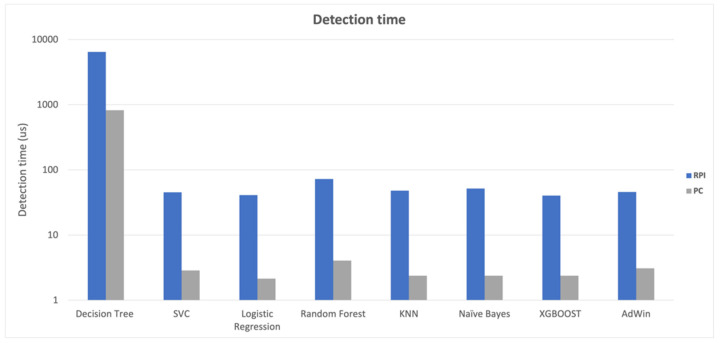
Detection time (in micro-seconds; µs) was higher in the Detection Tree algorithm than in all the others, with it always being lower in the PC-based (PC) than in Raspberry-based (RPI) gateways. Note: SVC—C-Support Vector Classification; KNN—K-Nearest Neighbors.

**Figure 7 animals-14-00938-f007:**
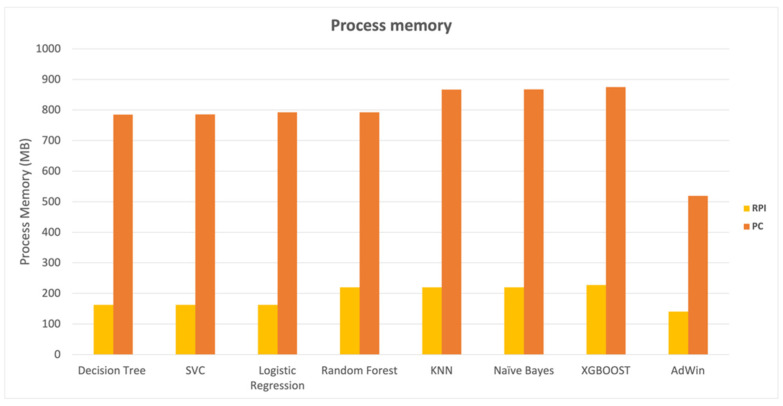
The process memory requirements were lower for all algorithms in Raspberry-based (RPI) than in the PC-based gateways. Note: SVC—C-Support Vector Classification; KNN—K-Nearest Neighbors.

**Figure 8 animals-14-00938-f008:**
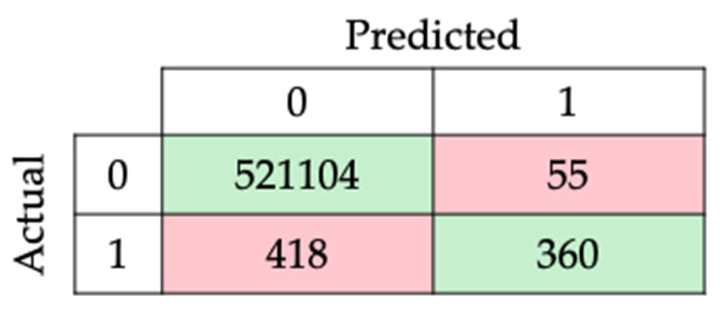
Confusion matrix of the best model for unchanged dataset.

**Figure 9 animals-14-00938-f009:**
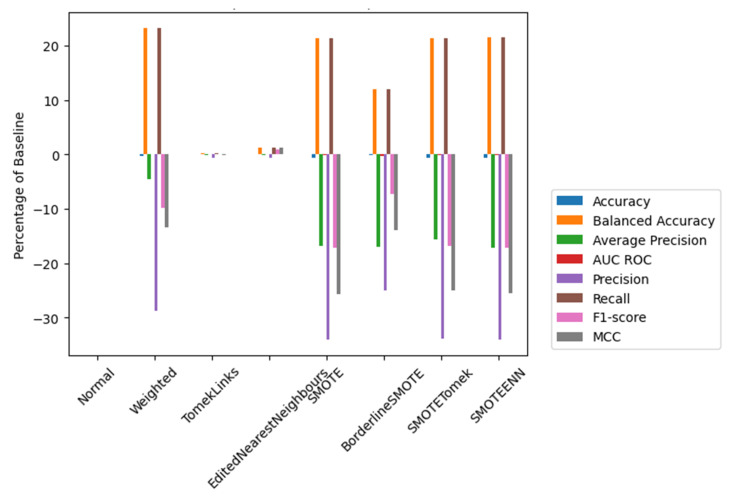
Technique performance score percentages compared to baseline.

**Figure 10 animals-14-00938-f010:**
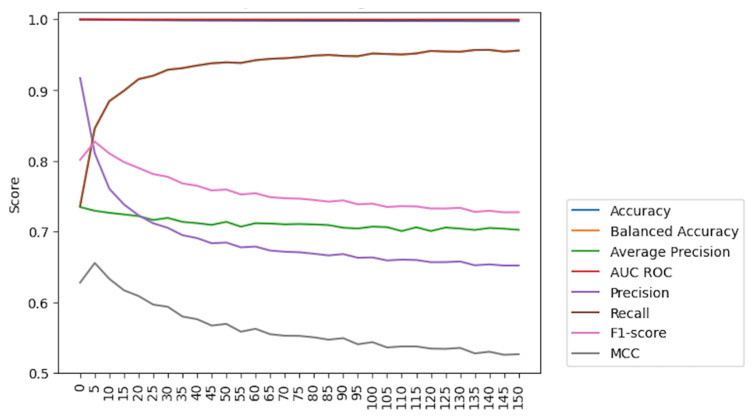
Weighted XGBoost performance evolution.

**Figure 11 animals-14-00938-f011:**
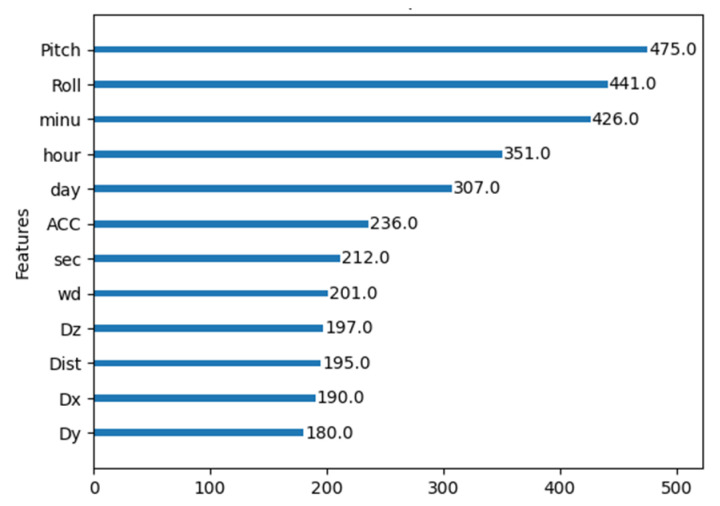
Pitch, Roll, and minute were the most important learning features.

**Figure 12 animals-14-00938-f012:**
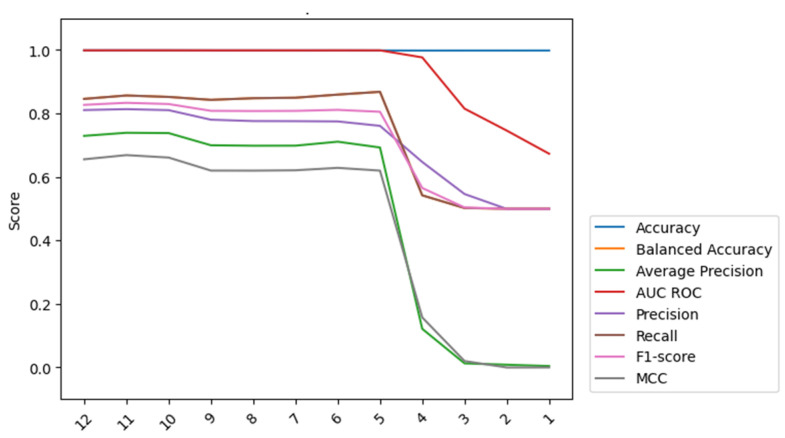
The model scores increased until the number of features in the model reached five.

**Figure 13 animals-14-00938-f013:**
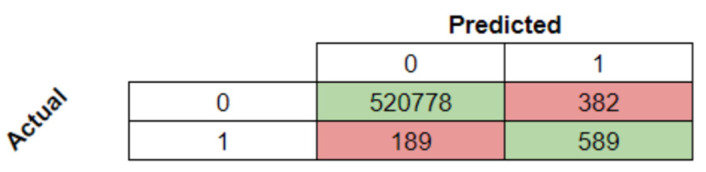
Confusion matrix of the best model.

**Table 1 animals-14-00938-t001:** Time classes created in the clustering process.

Class	Description
−5	5 h before
−4	4 h before
−3	3 h before
−2	2 h before
−1	1 h before
0	within kidding hour
1	1 h after

**Table 2 animals-14-00938-t002:** The number of concept changes was higher on the kidding day (P) than the day before (E).

Collar ID	Eve/Partum	Pitch	Activity
14	E	22	8
	P	38	18
	Incr. %	72.73	125.00
2	E	20	6
	P	40	14
	Incr. %	100.00	133.33
7	E	10	2
	P	21	6
	Incr. %	110.00	200.00
Average	E	17	5
	P	33	13
	Incr. %	94	160

**Table 3 animals-14-00938-t003:** The Decision Tree model presented the best precision results (f1-score, accuracy, and recall) of the various data classification algorithms in time classes 0 and 5.

	Time Classes						
Model	−5	−4	−3	−2	−1	0	+1
F1-score							
Decision Tree	0.3401	0	0	0.2612	0	0.5432	0
SVC	0.3617	0.1818	0.2017	0.2504	0.1106	0.5539	0.2164
Logistic Regression	0.2718	0.1981	0.0795	0.1538	0	0.5083	0.2508
Random Forest	0.3672	0.3506	0.2307	0.2749	0.1812	0.5787	0.2671
KNN	0.3754	0.3163	0.2333	0.2563	0.2214	0.5459	0.2759
Naïve Bayes	0.2787	0.2764	0.0614	0.1511	0.0462	0.5298	0.2500
XGBoost	0.3577	0.2361	0.2195	0.2441	0.1480	0.5539	0.26
Precision							
Decision Tree	0.2154	0	0	0.2203	0	0.4784	0
SVC	0.2773	0.3063	0.2134	0.2429	0.1523	0.4793	0.3548
Logistic Regression	0.2103	0.2477	0.2368	0.2105	0	0.3937	0.2237
Random Forest	0.3551	0.3591	0.2485	0.2791	0.1912	0.5448	0.2718
KNN	0.4076	0.3653	0.2671	0.2596	0.2222	0.4833	0.2477
Naïve Bayes	0.2080	0.2369	0.1765	0.2308	0.1290	0.4666	0.2630
XGBoost	0.2807	0.3075	0.2890	0.2579	0.1692	0.4812	0.3144
Recall							
Decision Tree	0.8080	0	0	0.3209	0	0.6282	0
SVC	0.5199	0.1293	0.1912	0.2585	0.0869	0.6559	0.1557
Logistic Regression	0.3841	0.1651	0.0478	0.1212	0	0.7170	0.2854
Random Forest	0.3750	0.3396	0.2265	0.2692	0.1737	0.6204	0.2642
KNN	0.3478	0.2788	0.2071	0.2531	0.2207	0.6271	0.3113
Naïve Bayes	0.4221	0.3318	0.0372	0.1123	0.0282	0.6127	0.2382
XGBoost	0.4928	0.1916	0.1770	0.2317	0.1315	0.6526	0.2217

Note: SVC—C-Support Vector Classification; KNN—K-Nearest Neighbors.

**Table 4 animals-14-00938-t004:** Precision, detection time and memory impact differed between the algorithms.

Model	Precision −4 h	Precision Kidding Hour	Detection Time (µs)	Memory Impact (MB)
Decision Tree	0	0.4784	6480	**162.75**
SVC	0.3063	0.4793	45.3	**162.76**
Logistic Regression	0.2477	0.3937	41.2	**162.76**
Random Forest	0.3591	**0.5448**	72.5	220.15
KNN	**0.3653**	0.4833	48.2	220.17
Naïve Bayes	0.2369	0.4666	51.7	219.95
XGBOOST	0.3075	0.4812	**40.5**	227.57

Note: SVC—C-Support Vector Classification; KNN—K-Nearest Neighbors. Bold values highlight the algorithms that perform better with respect to each parameter.

**Table 5 animals-14-00938-t005:** Classification report for unchanged dataset.

	Precision	Recall	F1-Score	Support
0	1.0	1.0	1.0	521,160
1	0.87	0.46	0.60	778
Accuracy			1.0	
Macro avg	0.93	0.73	0.80	521,938
Weighted avg	1.0	1.0	1.0	521,938

**Table 6 animals-14-00938-t006:** Scores best parameters after the balancing process.

	Score	Standard Deviation	Percentage Weighted 5	Percentage Baseline
Accuracy	0.999	0.000	+0.001	−0.013
Balanced Accuracy	0.879	0.037	+1.294	+14.275
Average Precision	0.760	0.037	+0.691	+2.479
AUC ROC	0.999	0.000	+0.009	+0.019
Precision	0.808	0.020	+1.294	−10.901
Recall	0.879	0.020	+0.050	+14.275
F1-score	0.840	0.019	+0.567	+3.839
MCC	0.684	0.037	+1.239	+5.571

**Table 7 animals-14-00938-t007:** Classification report for the balanced dataset.

	Precision	Recall	F1-Score	Support
0	1.0	1.0	1.0	521,160
1	0.61	0.76	0.67	778
Accuracy			1.0	
Macro avg	0.80		0.84	521,938
Weighted avg	1.0	1.0	1.0	521,938

## Data Availability

The dataset is publicly available at https://figshare.com/s/925215e8ea73da4b01f2, accessed on 1 March 2024.

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
