# Peer review of "Exploring the Potential of Machine Learning Algorithms Associated with the Use of Inertial Sensors for Goat Kidding Detection"

_animals, 2024, doi:10.3390/ani14060938_

Round 1

Reviewer 1 Report

Comments and Suggestions for Authors

Overall, the manuscript is well-written and well-structured. The methodology and results presented are sound. Please check below my specific comments.

1.  I suggest add 'goat' to the title.
2. On line 308, why a 1-h interval was used to create the classes? and why 5 hours before was decided to be the earliest class? Here I miss some references. 
3. On line 315, it seems a random splitting was used to generate the datasets for training and testing. This may have raised the issue of data leakage between two sets as the observations from the same goats can show up in both sets. A goat-level splitting may be better. Please clarify.
4. On line 373, I didn't see grey lines in Figure 2, I guess it should be 'green lines'.
5. On line 377, something is missing here 'the , where:'. Again, 'grey lines' should be corrected. Anyway this sentence seems to be useless as it only replicates the first sentence.
6. The same problem with the next sentence on lines 378-380. These sentences have to be rewritten to be concise and clear.
7. On line 385, here what do the authors mean by 'does' movements? Please modify.
8. On line 615, remove '['.
9. On line 666, here please check the use of 'ground distance'.
10. On line 673, Figure 8 and Figure 13.

Author Response

Dear reviewer 1,

We would like to take this opportunity to thank you for your review efforts, as well as your comments and suggestions.
We submit the revised version of the article, with the changes highlighted.

-----

Overall, the manuscript is well-written and well-structured. The methodology and results presented are sound. Please check below my specific comments.
We thank the reviewer for effort in reviewing the paper and for his valuable comments.

  1. I suggest add 'goat' to the title.

We added it.

  1. On line 308, why a 1-h interval was used to create the classes? and why 5 hours before was decided to be the earliest class? Here I miss some references. 
    The choice of time intervals was due to an expectation created by the analysis of related work carried out on sheep. Most of these studies indicate that signs of imminent birth are visible up to 12 hours before (e,g, [20], [22],[23] ), and also describe that the mother licks the goatling in the two hours after birth. The literature also reports that the offspring lasts a few minutes, something that we have verified in subsequent work in which we have also used video images. We did not use intervals of up to twelve hours because we were aware of the limited size of our dataset, and because we realized that this would make our prediction results even worse.

The inclusion of a class related to the time after birth was due to the fact that the behavior of small ruminants licks the offspring at the end of the process. And this event strongly marks a successful birth process, something that can be used to disarm the birth alarm, or notify the human guardian that everything went well with the goat.

  1. On line 315, it seems a random splitting was used to generate the datasets for training and testing. This may have raised the issue of data leakage between two sets as the observations from the same goats can show up in both sets. A goat-level splitting may be better. Please clarify.
    The division described in line 315 refers to the application of data classification algorithms, the 75 - 25% division can happen randomly, as the data is labeled by class. The classes already indicate 5h, 4h, 3h, etc... It is not the same situation regarding the application of algorithms to identify context changes. In this case, it makes perfect sense to align the data chronologically and analyze it in relation to the time scale.
  2. On line 373, I didn't see grey lines in Figure 2, I guess it should be 'green lines'.
    Yes reviewer is right, we changed it.
  3. On line 377, something is missing here 'the , where:'. Again, 'grey lines' should be corrected. Anyway this sentence seems to be useless as it only replicates the first sentence.
    Yes reviewer is right, we removed the two lines.
  4. The same problem with the next sentence on lines 378-380. These sentences have to be rewritten to be concise and clear.
    The reviewer was right, the sentences were edited to remove repetitions and make them clearer.
  5. On line 385, here what do the authors mean by 'does' movements? Please modify.
    It should be goats, we edited it.
  6. On line 615, remove '['.
    We did it.
  7. On line 666, here please check the use of 'ground distance'.
    We did it.
  8. On line 673, Figure 8 and Figure 13.

 We did it.

Reviewer 2 Report

Comments and Suggestions for Authors

1. The main question addressed by the Research was the creation of an algorithm based on animal learning models.

2. The paper referred to the use of these algorithms in goats, since Little work has been done in such specie.

3. In a general manner when compared to other published material, it ads quite a few, however, in relation to goats, it adds new knowledge.  Before this study, few Works have been done about goats learning or following patterns.  Also, it adds that with some adjustments to the methodology (limitation on the spread of the simples and behaviors analyzed) is possible to register changes in patterns of movement and This, making possible to create some algorithms explaining changes in goats behaviour previous to parturition

4. Regarding the methodology, it has more observations, to analyze a greater flock.

5. The conclusions are consistent with the evidence;   however, these are too large.

In that sense, my conclusion would have been.

“All algorithms proved feasibility for implementation on a microcomputer in terms of computational demands and execution time, enabling real-time and edge detection.

Additionally, the majority of algorithms took less than 50 microseconds to complete the detection process, allowing to trigger alarms requesting 696 prompt human assistance when needed.

The effort to identify kidding day resulted in a notable enhancement in the algorithm's effectiveness, owing to the reduced number of classes it had to understand. Nevertheless, the outcomes remain unsatisfactory since the detection of kidding day was achieved with only 61% accuracy.”

6. About the references, I think references are appropriate. However, references could be improved and loook for more recent referfences since some are date back 2019 up to 2022. I only would recommend to search for some more recent references between the uear 2022 and 2023. The rest are ok and quite recent as well.

7. Conclusion must be more precise and concise. Two or three lines for each main conclusions is enough, the rest could be added to the discussion

Author Response

   Dear reviewer 2,

   We would like to take this opportunity to thank you for your review efforts, as well as your comments and suggestions.
We submit the revised version of the article, with the changes highlighted.

Comments and Suggestions for Authors

  1. The main question addressed by the Research was the creation of an algorithm based on animal learning models.

  2. The paper referred to the use of these algorithms in goats, since Little work has been done in such specie.
  3. In a general manner when compared to other published material, it ads quite a few, however, in relation to goats, it adds new knowledge.  Before this study, few Works have been done about goats learning or following patterns.  Also, it adds that with some adjustments to the methodology (limitation on the spread of the simples and behaviors analyzed) is possible to register changes in patterns of movement and This, making possible to create some algorithms explaining changes in goats behaviour previous to parturition
  4. Regarding the methodology, it has more observations, to analyze a greater flock.
  5. The conclusions are consistent with the evidence;   however, these are too large.

In that sense, my conclusion would have been.

“All algorithms proved feasibility for implementation on a microcomputer in terms of computational demands and execution time, enabling real-time and edge detection.

Additionally, the majority of algorithms took less than 50 microseconds to complete the detection process, allowing to trigger alarms requesting 696 prompt human assistance when needed.

The effort to identify kidding day resulted in a notable enhancement in the algorithm's effectiveness, owing to the reduced number of classes it had to understand. Nevertheless, the outcomes remain unsatisfactory since the detection of kidding day was achieved with only 61% accuracy.”

We thank the reviewer for effort in reviewing the paper and for his comments, and for the help in summarizing conclusions. We edited the text of the conclusions to summarize them in accordance with the suggestion you presented.

  1. About the references, I think references are appropriate. However, references could be improved and loook for more recent referfences since some are date back 2019 up to 2022. I only would recommend to search for some more recent references between the uear 2022 and 2023. The rest are ok and quite recent as well.

There is still no work related to detecting goat kidding. We agree with the reviewer's opinion regarding the importance of updating the related work and presenting recent references, but in terms of goat birth detection there are no other references besides the dataset paper.

  1. Conclusion must be more precise and concise. Two or three lines for each main conclusions is enough, the rest could be added to the discussion

We follow your suggestion, thank you very much.

Reviewer 3 Report

Comments and Suggestions for Authors

You have identified a number of approaches to improve the accuracy of detection.

L610: "they are promising in that increasing the number of samples should allow for significant improvements in the learning model "

L633: "...can be improved by adding GPS trackers to the accelerometer devices..."

L644: "...pitch and roll angles, as well as temporal features, ranked among the top five most impactful

L698: "The effort to identify kidding day resulted in a notable enhancement in the algorithm's effectiveness, owing to the reduced number of classes it had to understand."

Which of these would you recommend as having the most probability of increasing accuracy?  if you had a larger sample size would that get you the accuracy you need?  What is the estimated potential of adding GPS? How do these compare to improved algorithms?

This is not essential to include in the paper but they are questions that arose when I read it.  If you could opine more specifically on directions for improvement it would add interest to the paper.

play 

Author Response

Dear reviewer 3,

We would like to take this opportunity to thank you for your review efforts, as well as your comments and suggestions.
We submit the revised version of the article, with the changes highlighted.

Comments and Suggestions for Authors

You have identified a number of approaches to improve the accuracy of detection.

L610: "they are promising in that increasing the number of samples should allow for significant improvements in the learning model "

Reviewer is right, we did the sentence to make it clear.

L633: "...can be improved by adding GPS trackers to the accelerometer devices..."

We decided to remove it, especially because GPS receivers have limitations to be used indoors.

L644: "...pitch and roll angles, as well as temporal features, ranked among the top five most impactful

We did it.

L698: "The effort to identify kidding day resulted in a notable enhancement in the algorithm's effectiveness, owing to the reduced number of classes it had to understand."

We edited it in order to make it clear.

Which of these would you recommend as having the most probability of increasing accuracy?  if you had a larger sample size would that get you the accuracy you need?  What is the estimated potential of adding GPS? How do these compare to improved algorithms?

We tried some approaches to improve accuracy, some before carrying out this work, others after a new monitoring test. From these experiments we confirmed that the classification of records into time classes was a successful strategy and that the results improve significantly with the increase in the dataset. In this sense, we are enriching the dataset using a higher sampling frequency and a larger herd, in order to allow us to preserve the diversity of behaviors in the learning model. We never considered the use of GPS due to limitations of indoor operation, until we found a proposal for detecting sheep births (i.e. [22]) that uses it and reports improvements in results. In fact, we believe it may have a contribution, but limited. As we wrote above, we decided to remove GPS receivers from future work proposals.

This is not essential to include in the paper but they are questions that arose when I read it.  If you could opine more specifically on directions for improvement it would add interest to the paper.

 Thank you.

Round 2

Reviewer 1 Report

Comments and Suggestions for Authors

The authors have answered my questions.